

# Substrate type and light intensity determine *lampenflora* concentration on paleontological remains in show caves

Elena Piano[1,2], Marta Zunino[3,4], Giuseppe Nicolosi[1,5], Isabella Nicole Pisoni[6], Alice Cimenti[1], Alberto Cina[6] and Marco Isaia[1,2]

[1] Department of Life Sciences and Systems Biology, University of Turin, Torino, Italy
[2] National Biodiversity Future Center, Palermo, Italy
[3] Grotte di Toirano, Toirano, Italy
[4] Soprintendenza Archeologia, Belle Arti e Paesaggio per le province di Alessandria, Asti e Cuneo, Alessandria, Italy
[5] CNR - Istituto di Ricerca sulle Acque, Verbania, Italy
[6] Polytechnic Institute of Turin, Torino, Italy

Corresponding author
Elena Piano, elena.piano@unito.it

## ABSTRACT

Artificial lighting in show caves is responsible for the growth of nuisance photosynthetic organisms, the so-called *lampenflora*, causing aesthetic, chemical and physical damage to cave cultural heritage, including paleontological resources *in situ*. This study focuses on the role of substrate in determining the concentration of *lampenflora* on paleontological findings in show caves, using the bone deposit "Cimitero degli Orsi" in the Toirano show cave (NW-Italy) as a testing ground. Specifically, we investigated whether the concentration of three distinct photosynthetic microorganisms—cyanobacteria, diatoms, and green algae—varies on different substrates, *i.e.*, bones, rock, and soil, also keeping into account the role of light intensity. Our findings revealed that, among the tested organisms, diatoms exhibited higher concentration on bones compared to other substrates and it was even higher at increasing light intensity. On the other hand, cyanobacteria increased their concentration at increasing light without a clear preference for a specific substrate, while the presence of green algae was higher on rock and soil substrates rather than bones. When modelling the concentration of photosynthetic microorganisms within the bone deposit under different scenarios of light intensity reduction, we predicted a general decrease of all groups, that was stronger in cyanobacteria and green algae and weaker in diatoms on bone substrates. These results provide valuable insights on the colonization of nuisance photosynthetic microorganisms on bone substrates exposed to artificial lighting, with management implications for the conservation of paleontological findings in show caves.

## INTRODUCTION

Caves have attracted humankind since prehistoric times, being used as shelters, for rituals, or for food conservation (*Mulec, 2014*). Due to their peculiar habitat conditions of spatial confinement, absence of light and climatic stability (*Bastian & Alabouvette, 2009*), caves are

extraordinary habitats for the preservation of ancient traces of human frequentation and fossils (*Romano et al., 2019*). Accordingly, many caves with extraordinary paleontological remains have been converted into potent tourist attractions (*Piano et al., 2022*; *Sanna, Chiarini & De Waele, 2023*).

Since the first evidence of cave visitors in 1213 in the Postojna Cave (Slovenia), tourism in caves grew considerably, leading to the conversion of natural caves into the so-called show caves—caves made accessible to the public for touristic purposes, managed by a government or commercial organization, where paying visitors experience the cave *via* constructed trails, guided tours, artificial lights, and regular opening hours (*Cigna, 2016*). Such tourist attractions may reach impressive numbers of visitors (up to 500,000/year/cave) and may profit up to 3 billion US $ per year (*Chiarini, Duckeck & De Waele, 2022*).

The conversion of wild caves into tourist attractions and the associated human transformations significantly contribute to the alteration of the natural stability of the subterranean ecosystem (*Piano et al., 2024*). More in detail, the cave equilibrium is highly compromised as a massive amount of additional energy is introduced due to the presence of visitors and the installation of artificial lights (*Cigna, 2016*). Visitors release energy in form of heat and $CO_2$ that alter the subterranean microclimate (*Novas et al., 2017*; *Addesso, Bellino & Baldantoni, 2022a*). These environmental changes may in turn affect air quality (*Addesso et al., 2022b*) and the geochemical balance of speleothem formation, influencing condensation-corrosion phenomena (*Fernandez-Cortes et al., 2011*; *Addesso et al., 2023a*). Furthermore, human visitors carry microbial propagules (*Porca et al., 2011*; *Mammola et al., 2017*; *Alonso et al., 2019*; *Biagioli et al., 2023*; *Piano et al., 2023a*) and organic matter inside caves (*Kukla et al., 2018*) with dramatic consequences on all ecosystem components (*Piano et al., 2024*).

Besides environmental impacts, tourism in caves may also have direct and indirect impacts on the resident paleo-archaeological heritage, especially when archaeological remains and/or paleontological deposits are open to visitors (*Bontemps et al., 2022*). One of the major threats is represented by the installation of artificial lights which determines the establishment and proliferation of photosynthetic organisms, the so-called *lampenflora*, mostly constituted by diatoms, cyanobacteria, and green algae, exploiting the light produced by artificial lamps (*Mulec, 2019*; *Baquedano Estevez et al., 2019*). *Lampenflora* colonization of cave surfaces may cause alterations of substrates, which comprehend aesthetic damage (*Mulec, 2019*), chemical deterioration (*Baquedano Estevez et al., 2019*) and physical alteration (*Lian, Yuan & Liu, 2011*; *Cuezva et al., 2012*), ultimately jeopardizing the touristic value of show caves. For this reason, *lampenflora* has been extensively studied in literature, with research focusing either on ecological drivers (*e.g.*, *Borderie et al., 2014*; *Piano et al., 2015*; *Piano, Nicolosi & Isaia, 2021*), algal and bacterial composition of photosynthetic biofilms (*e.g.*, *Alonso et al., 2019*; *Miller et al., 2009*; *Miller et al., 2012*; *Pfendler et al., 2018*) or management and eradication strategies (*e.g.*, *Addesso et al., 2023b*; *Havlena et al., 2021*; *Pfendler et al., 2017*).

The development of photosynthetic biofilms on cave surfaces may also represent a major threat for the conservation of paleo-archaeological heritage, with possible economic repercussions for the local managing authorities (*Bontemps et al., 2022*). Consequences on

their conservation can sometimes be dramatic and may even lead to the closure of the show cave, as happened in 1963 to the Lascaux cave (France), one of the most famous caves worldwide for its extraordinary Palaeolithic paintings (*Bastian & Alabouvette, 2009*; *Bastian et al., 2010*).

Previous studies demonstrated that several parameters related to artificial lights may influence the concentration of the photobiota in show caves, including light intensity (*Piano et al., 2015*), light duration (*Borderie et al., 2014*; *Piano, Nicolosi & Isaia, 2021*), and light type (*Havlena et al., 2021*). Besides this, evidence in literature suggests that the concentration of photosynthetic biofilms may also vary depending on the substrate, *e.g.*, *Miller et al. (2009)* and *Miller et al. (2012)* highlighted that different substrates may be colonized by different microorganisms. While several studies examined the differential concentration of photosynthetic microorganisms on building materials (see *Sanmartín et al., 2021* for a review on this topic), and their biodeterioration on cultural heritage, such as catacombs or fossil deposits (*e.g.*, *Gaylarde et al., 2011*; *Marano et al., 2016*), this aspect has been poorly investigated on the paleontological heritage of show caves.

We here estimated the *lampenflora* colonization of different substrates within the 'Cimitero degli Orsi' bone deposit of the Toirano show cave (NW-Italy). Our aim was to understand whether fossil bones were differentially colonized by different photosynthetic microorganisms focusing on the three main groups commonly colonizing cave walls and speleothems in show caves—cyanobacteria, diatoms, and green algae. We measured their concentration using a pulse amplitude modulated (PAM) fluorimeter that provides an instantaneous and *in situ* estimation of the chlorophyll-*a* (chl-*a*) concentration of these three benthic photosynthetic organisms, with a standard methodology that could be easily replicated in other caves providing comparable results. These measures were used as proxies to evaluate whether these groups exhibited different concentrations in response to different substrates, namely bones, rock and soil. Given the influential role of light intensity on the concentration of photosynthetic microorganisms in show caves (*Piano et al., 2015*; *Piano, Nicolosi & Isaia, 2021*; *Piano, Nicolosi & Isaia, 2023b*), we also examined their response to this parameter. Specifically we hypothesized that: (i) light intensity favours all examined groups; and (ii) the three groups display different concentrations on the three examined substrates, underlying a differential substrate preference. Accordingly, we used these results to predict the concentration of photosynthetic microorganisms under different scenarios of light intensity, aiming at providing suggestions for the management and conservation of the paleo-archaeological heritage in show caves.

## MATERIALS & METHODS

### Sampling area

We performed our study in the Toirano show cave (NW-Italy), which is part of the Toirano karst system. This system is located on the Briançonnais domain of the Ligurian Alps and the *San Pietro dei Monti* formation, which mainly comprises Middle Triassic dolomitic limestones (*Boni et al., 1971*; *Cavallo, 2001*; *Colombu et al., 2021*). The Toirano karst system extends for three km and is composed of four different caves, but only two of them are open

to the public, namely the Santa Lucia Inferiore Cave ([Li 59], 8.2032°E, 44.1344°N; 201 m a.s.l., 778 m long) and the Bàsura Cave ([Li 55], 8.2020°E, 44.1378°N; 186 m a.s.l., 890 m long), forming the Toirano show cave. The morphology of the two caves suggests that they were generated by slow flowing water following an ascending path and subsequently shaped by condensation-corrosion phenomena due to the convection of external warm and wet air masses, and vapours produced by bats and decay of guano deposits (*Colombu et al., 2021*). The Toirano show cave was opened to the public in 1953 and it is visited by an average number of 75,000 visitors per year (data provided from cave managers referring to 2019). The average cave temperature is around 16 °C (average measure obtained with Thermochrone iButtons® left in the caves from June 2020 to February 2021). The tourist path is almost 1,300 m long, including the artificial tunnel connecting Bàsura Cave and Santa Lucia Inferiore Cave, and it is equipped with a manual step-by-step illumination system, constituted by incandescent lights, which are progressively turned on and off during visits. Incandescent lamps emit a continuous spectrum of light extending from about 300 nanometers, in the ultraviolet region, to about 1,400 nanometers, in the near infrared region. Both caves are sealed with compartment doors. The main attraction is represented by the exceptional prehistoric and paleontological heritage of the Bàsura Cave, consisting of traces of human activities, especially footprints, dating back to 14 ky cal. BP, and a cave bear (*Ursus spelaeus*) bone deposit (the so-called 'Cimitero degli Orsi', Fig. 1) dated about between 50,000 years and 28,000 years ago (*Romano et al., 2019*; *Zunino et al., 2022*). The 'Cimitero degli Orsi' is a remarkable bone bed composed of thousands of bear bones located about 400 m from the entrance of the cave. This bone deposit was discovered and partially excavated in 1951, during which a small part of the remains was removed and most of the bones were left *in situ* as a tourist attraction on the tour route (*Zunino et al., 2022*). These bones are embedded within a soil displaying a groundmass with an open porphyric distribution pattern and a micromass of yellow argillo-phosphatic sediment containing cryptocrystalline phosphate and Fe–Mn oxides. The coarse fraction was mainly sandy or silty angular quartz and mica grains, poorly sorted and arranged in bands, with some rounded clasts, with grains included polycrystalline quartz, mica schist, rock fragments, quartz, and pyroxene (for further details on soil composition; see *Zunino et al., 2022*).

## Data collection

Data collection on the field (File S1) was performed following the authorization provided by the "Soprintendenza Archeologica, Belle Arti e Paesaggio" (File S2). We identified 60 sampling plots within the deposit, consisting of a circle of 20 cm diameter, distributed across three different types of substrate (Fig. 2), *i.e.,* bare soil (Soil = 18 sites)), rock (Rock = 20 sites) and bones (Bone = 22 sites). Data were collected as previously described in *Piano et al. (2015)*. In each sampling plot, we measured three replicates of the concentration of the three main photosynthetic groups constituting the photobiota in caves, namely cyanobacteria, diatoms, and green algae. Measurements were conducted using the BenthoTorch® (BBE Moldaenke GmbH, Schwentinental, Germany), a PAM fluorimeter that emits light at three distinct wavelengths (470, 525, and 610 nm) and

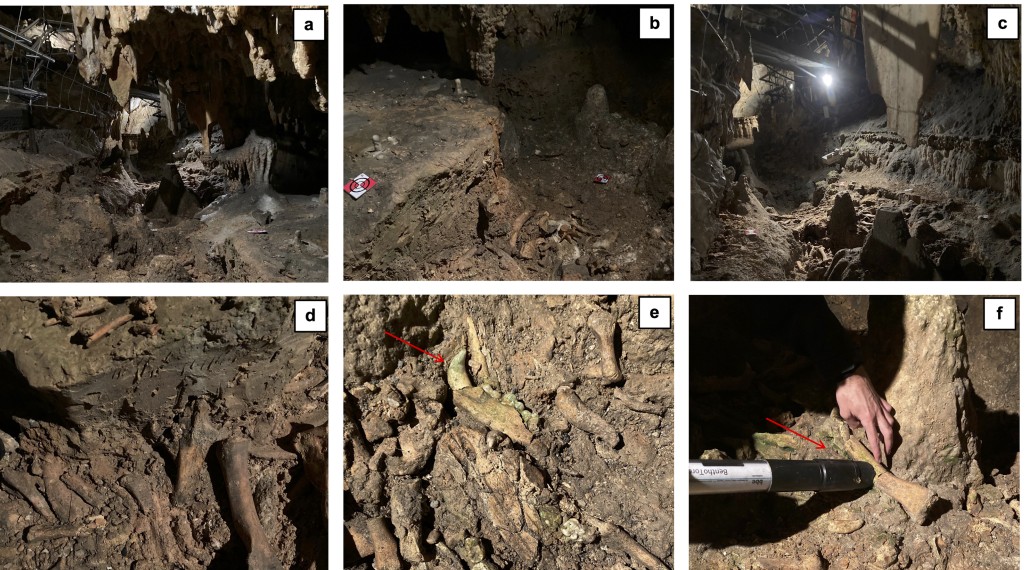

**Figure 1 Overview of the bone deposit 'Cimitero degli Orsi' (A–C), details of paleontological remains (D–E), and lampenflora measurements on fossil remains (F).** Red arrows indicate areas of aesthetic damage caused by photosynthetic microorganisms.

records the response of cyanobacteria, diatoms, and green algae at 690 nm. The device, equipped with an internal algorithm, provides an instantaneous and *in situ* estimation of the chl-*a* concentration of these three benthic photosynthetic organisms (μg chl-*a*/cm$^2$). Photosynthetic microorganisms are distinguished based on specific pigments characteristic of each of them, namely phycocyanin for cyanobacteria, chlorophyll-*c* and xanthophylls for diatoms, and chlorophyll-*b* for green algae. A 700 nm LED compensates for background reflectance (*Carpentier et al., 2013*). It is important to note that BenthoTorch® measurements should not are interpreted as direct assessments of algal biomass or biovolume, but rather as proxies for biofilm primary production (*Kahlert & McKie, 2014*). Additionally, measurements taken on thick biofilm mats or at chlorophyll-*a* levels above five μg/cm$^2$ may be skewed when compared to standardized laboratory techniques (*Echenique-Subiabre et al., 2016*; *Kaylor et al., 2018*; *Rosero-López et al., 2021*). It is also recommended that BenthoTorch® chlorophyll-*a* measurements be performed after a low-light adaptation period of 20–30 min, avoiding full sunlight exposure (*Kaylor et al., 2018*). However, these limitations are mitigated in cave environments due to the following conditions: (i) biofilms are very thin (<one mm); (ii) chlorophyll-*a* values are generally low (<one μg/cm$^2$); and (iii) biofilms are consistently subjected to low light levels. Consequently, we considered the measurements obtained with the BenthoTorch® to accurately represent the real concentration of the three targeted groups. It should be noted this method did not allow us to detect fossilized microorganisms that can be present on fossil bones (*e.g.*, *Stone & Yost, 2020*; *de Sousa et al., 2024*). However, we were interested in estimating the colonization of fossil bones by living photosynthetic microorganisms that are able to proliferate and thus damage the paleontological heritage of the cave. For this

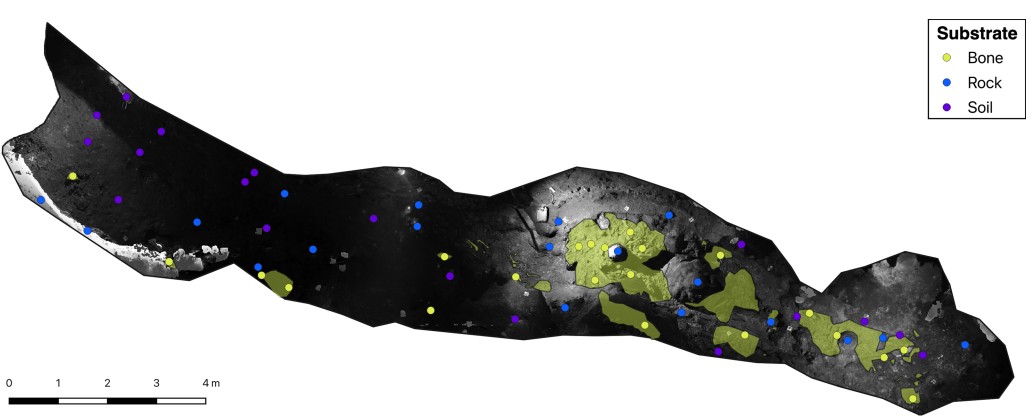

**Figure 2** Greyscale orthophoto of the 'Cimitero degli Orsi' with representation of sampling points and of bone deposits (yellow areas).

reason, we adopted a method that only measures active photosynthetic pigments, therefore providing an estimate of living organisms.

Besides considering substrate differences, we measured the intensity of incident light with a DELTA OHM S.r.l. LP 471 PHOT probe in each plot to keep into account the role of light intensity on the concentration of the three photosynthetic microorganisms.

## Statistical analysis

All statistical analyses were performed with the R statistical software (*R Core Team, 2023*).

For each photosynthetic group, we retained the median value of the three chl-*a* replicates collected in each plot to be included in the subsequent statistical analyses. We preferred the median over the mean value because it allowed us to obtain results unbiased by extreme values (*Legendre & Legendre, 2012*) that often occur in the measurements of *lampenflora* concentration (*Piano, Nicolosi & Isaia, 2023b*). We therefore retained the chl-*a* concentration of cyanobacteria, diatoms, and green algae as dependent variables for a total of three dependent variables.

Before proceeding with the statistical analyses, we performed data exploration by checking the distribution of both dependent and independent variables and identifying possible outliers or non-linear distributions of variables. A log-transformation was applied to light intensity to homogenize its distribution. Regarding dependent variables, we converted the green algae concentration into a binomial variable by assigning 1 to plots where the median value of green algae concentration was >0 due to the high zero-inflation. Before analyzing the response of photosynthetic organisms to environmental parameters, we checked whether plots located on different substrate types were exposed to comparable light intensities by performing a gaussian regression with log-transformed values of light intensity modelled against the substrate type. Based on these results, we then performed two sets of models: (i) a first set including all data to test the response of dependent variables to all examined environmental parameters (full dataset); and (ii) a second set including only plots under the threshold of 25 lux aiming at discriminating the role of substrates independently from light (restricted dataset). This threshold was obtained by

comparing the range of light intensity experienced by the three substrates and selecting the most restrictive one (*i.e.,* <25 lux). Specifically, the restricted dataset comprises only plots where all three substrate types are subjected to the same range of light intensity. This approach allowed us to better discriminate the substrate effect on the concentration of the three groups of photosynthetic microorganisms, minimizing confounding effects caused by variations in light intensity. Based on this approach, 12 plots were excluded for a total of 48 sampling plots retained (bare soil = 18 plots; rock = 15 plots; bones = 15 plots). Overall, we performed two models for each photosynthetic group, one with the full dataset and one with the restricted dataset, for a total of six models.

Given the absence of non-linear distributions of variables observed during the data exploration, we tested the predictor variables against the dependent variables, *i.e.,* light intensity, and substrate type, *via* generalized linear models (GLMs). For each dependent variable we then performed model validation with the function 'check_model' of the *performance* package (*Lüdecke et al., 2021*), which allowed us to verify the residual distribution, the presence of influential observations and the homogeneity of variance. Considering that our dependent variables did not assume negative values, we adopted a Gamma distribution—which was preferred over the Gaussian distribution, which allows prediction of negative values—with log link function for the chl-*a* concentration of diatoms and cyanobacteria, while we ran the binomial model using a complementary log–log link function (clog–log)—as recommended in *Zuur et al. (2009)* for datasets with unbalanced set of zeros (absences) and ones (presences)—for the presence/absence of green algae.

## Light intensity scenarios

Results of statistical models were used to perform *lampenflora* projections under the present scenario of illumination and under two alternative scenarios of light intensity reduction. Despite there are currently no immediate plans to replace the lighting system in the cave, it seems plausible that, in the near future, incandescent lights will eventually be substituted with LEDs in accordance with the guidelines provided by the International Association of Show Caves (*International Show Cave Association, 2014*). Given the ability to modulate light intensity in LEDs, we propose two potential scenarios for future light intensity reduction and estimate the resulting concentration decreases of the three photosynthetic groups. Based on this, we predicted abundance of cyanobacteria and diatom and presence/absence of green algae under the present light intensity and under −30% and −50% of light intensity compared to present.

Data on the present level of illumination of the whole deposit were obtained from an orthophoto in grayscale (Fig. 2). To obtain a greyscale orthophoto, we captured images utilizing a NIKON 800 camera equipped with a 36-megapixel sensor and a 20 mm focal length. The camera was set with a fixed shutter speed of 1/40 and an aperture of f/3.5, potentially resulting in image overexposure or underexposure depending on ambient lighting conditions. The orthophoto was obtained with a resolution of approximately 1 millimeter. Although the camera records reflected light, which is influenced by several features such as surface inclination and light absorption rather than incident light,

**Table 1** Mean values, standard deviations, and ranges of the chl-*a* concentration of the three examined photosynthetic groups and light intensity on the three substrates (*n* = 60).

| | | Bones | Rock | Soil |
|---|---|---|---|---|
| **Cyanobacteria** **(μg/cm²)** | Mean (±SD) | 0.21 (± 0.21) | 0.21 (± 0.35) | 0.06 (± 0.04) |
| | Range | 0.01–0.86 | 0.00–1.17 | 0.01–0.15 |
| **Diatoms** **(μg/cm²)** | Mean (±SD) | 0.36 (± 0.14) | 0.28 (± 0.36) | 0.14 (± 0.06) |
| | Range | 0.15–0.64 | 0.05–1.32 | 0.05–0.25 |
| **Green algae** **(μg/cm²)** | Mean (±SD) | 0.07 (± 0.23) | 0.47 (± 1.04) | 0.01 (± 0.01) |
| | Range | 0.00–0.90 | 0.00–3.59 | 0.00–0.03 |
| **Light intensity** **(lux)** | Mean (±SD) | 22.8 (± 24.9) | 14.4 (± 16.6) | 6.05 (± 6.50) |
| | Range | 0.44–91.3 | 0.40–63.7 | 0.37–24.4 |

we assumed the value of white for each sampling site as an indicator of the prevailing illumination characteristics.

We then distributed our sampling points on the greyscale orthophoto, and values of the reflected light were obtained by extracting the value of white with the function 'extract' of the *raster* package (*Hijmans, 2023*). To verify whether these values could be used as a proxy of light intensity, we correlated them with the light intensity values obtained in the field. As the correlation among the two parameters was high (Pearson correlation test: $r = 0.78$; *p*-value $< 0.001$), we then proceeded with *lampenflora* projections using the greyscale orthophoto as a proxy of light conditions.

*Lampenflora* projections under the present light scenario were obtained in R by applying the resulting equations of statistical models to the orthophoto with sampling points located on its surface. To obtain *lampenflora* projections under different light intensity scenarios, the greyscale orthophoto was divided by 1.5 and by 2 to simulate a 30% and 50% light intensity reduction, respectively. As statistical models indicated a significant variation in *lampenflora* concentration on bones for some photosynthetic groups, predictions on bones were performed by applying the specific equation to the raster of the bone substrate within the deposit. This raster was obtained with the function 'rasterize' of the *raster* package (*Hijmans, 2023*) applied to a shapefile obtained in QGIS (*QGIS.org, 2023*) by retracing the bones' contours from an RGB orthophoto with resolution of 1 mm. Overall, we obtained nine rasters projecting *lampenflora* concentration in the bone deposit, one for each photosynthetic group in each light intensity scenario.

## RESULTS

According to our data, diatoms and cyanobacteria were the dominant photosynthetic groups, being present in 58 and 59 out of 60 sampling plots, respectively. Conversely, green algae were less represented, being present only in 13 out of 60 plots. When considering the chl-*a* concentrations of the three examined photosynthetic groups, diatoms showed, on average, higher chl-*a* concentrations compared to cyanobacteria on all types of substrates, while green algae showed the highest concentration values, with extreme values on rock substrates (Table 1). Light intensity ranged from 0.37 to 91.33 lux (mean $= 15.0 \pm 19.2$ lux) (Table 1).

The regression analysis testing light intensity on different substrates showed that plots on soil were exposed to significantly lower values than those on bones ($t = -2.65$, $P = 0.010$), indicating that sampling sites located on bones are more illuminated than those on soil.

The results of the GLM performed on the full dataset of cyanobacteria (Table 2A) showed that the concentration of this photosynthetic group is not affected by the substrate type (Fig. 3A) while it showed a significant increase at higher values of light intensity (Fig. 3B). When testing the response of cyanobacteria in the restricted dataset, we obtained the same pattern (Table 2A), with no effect of substrate type (Fig. 4A) and a positive effect of light intensity (Fig. 4B). Conversely, according to the results of the GLMs performed on diatom concentration in the full dataset (Table 2A), substrate type emerged as a key factor, with diatom concentration being significantly higher on bone substrates compared to the other substrate types here examined, *i.e.,* rock and soil (Fig. 3A). Diatom concentration was also significantly affected by light intensity, showing increasing values at increasing light intensity (Fig. 3C). When testing the diatom response by controlling for the confounding effect of light intensity, *i.e.,* using the reduced dataset (Table 2B), the results confirmed the significant effect of substrate type, with higher values of diatom concentration on bones than on rock and soil (Fig. 4A), while we could not detect any significant effect of light intensity (Fig. 4C). GLMs performed on the presence/absence of green algae by using the full dataset (Table 2A) demonstrated that the probability of presence of this photosynthetic group is significantly lower on bones than on rock and soil (Fig. 3A), but it is significantly favoured by increasing light intensity, especially at values > 3 log-transformed lux, which correspond to approximately 20 lux (Fig. 3D). When controlling for the confounding effect of light intensity on the different substrate types, the results of the GLM (Table 2B) showed that the effect of the substrate type was not significant (Fig. 4A), but the light intensity significantly increases the probability of presence of green algae especially at values > 3 log-transformed lux, which correspond to approximately 20 lux (Fig. 4D).

*Lampenflora* projections (Fig. 5) demonstrated that the area with the highest presence of bones within the bone deposit is highly colonized by all photosynthetic microorganisms. The concentration of cyanobacteria and the probability of presence of green algae are expected to decrease under the −30% light intensity reduction scenario and even more under the −50% light intensity reduction scenario. However, such reduction is expected to be much smaller for diatoms.

## DISCUSSION

When examining the main environmental drivers of *lampenflora* concentration within the bone deposit "Cimitero degli Orsi" in the Toirano show cave, we revealed a general positive correlation of the concentration of the three examined photosynthetic groups, *i.e.,* cyanobacteria, diatoms and green algae, with increasing values of light intensity. This result strengthens the acknowledged role of artificial light in enhancing the concentration of *lampenflora* in show caves. Despite photosynthetic species colonizing cave walls and speleothems demonstrated to survive long periods of darkness (*Piano, Nicolosi & Isaia, 2023b*), evidence in literature highlights that light intensity (*Roldán et al., 2004*; *Borderie*
**Table 2   Results of the GLMs analysis for the three examined photosynthetic groups in relation to light intensity (Light_intensity) and substrate type (Substrate) performed on the (a) full dataset, and (b) restricted dataset.** Estimated parameters ($\beta$-est), standard errors (SE), $z$-values ($z$) and $p$-values ($P$) for each covariate retained in the final model are reported. The category "Bone" represents the reference level for the categorical variable Substrate thus results refer to differences of rock (Rock_substrate) and soil (Soil_substrate) from Bone. Significant results are reported in bold.

| | | $\beta$-est ($\pm$SE) | t | P |
|---|---|---|---|---|
| **(a) Full dataset** | | | | |
| **Cyanobacteria** ($n = 59$) | Light_intensity (lux) | 0.527 ($\pm$0.139) | 3.79 | **<0.001** |
| | Rock_substrate | 0.522 ($\pm$0.381) | 1.37 | 0.176 |
| | Soil_substrate | −0.303 ($\pm$0.405) | −0.748 | 0.458 |
| **Diatoms** ($n = 58$) | Light_intensity (lux) | 0.223 ($\pm$0.068) | 3.26 | **0.002** |
| | Rock_substrate | −0.632 ($\pm$0.155) | −4.08 | **<0.001** |
| | Soil_substrate | −0.759 ($\pm$0.154) | −4.94 | **<0.001** |
| **Green algae** ($n = 60$) | Light_intensity (lux) | 1.66 ($\pm$0.458) | 3.62 | **<0.001** |
| | Rock_substrate | 1.96 ($\pm$0.834) | 2.35 | **0.019** |
| | Soil_substrate | 2.66 ($\pm$1.03) | 2.58 | **0.010** |
| **(b) Restricted dataset** | | | | |
| **Cyanobacteria** ($n = 47$) | Light_intensity (lux) | 0.485 ($\pm$0.189) | 2.56 | **0.014** |
| | Rock_substrate | −0.339 ($\pm$0.432) | −0.784 | 0.437 |
| | Soil_substrate | −0.230 ($\pm$0.410) | −0.561 | 0.578 |
| **Diatoms** ($n = 46$) | Light_intensity (lux) | 0.168 ($\pm$0.085) | 1.97 | 0.055 |
| | Rock_substrate | −0.613 ($\pm$0.196) | −3.13 | **0.003** |
| | Soil_substrate | −0.634 ($\pm$0.189) | −3.36 | **0.002** |
| **Green algae** ($n = 48$) | Light_intensity (lux) | 1.58 ($\pm$0.652) | 2.42 | **0.016** |
| | Rock_substrate | 1.07 ($\pm$1.24) | 0.86 | 0.389 |
| | Soil_substrate | 1.85 ($\pm$1.15) | 1.61 | 0.106 |

*et al., 2014*; *Piano et al., 2015*; *Piano, Nicolosi & Isaia, 2021*), light duration (*Planina, 1974*; *Piano, Nicolosi & Isaia, 2021*) and light type (*Roldán et al., 2006*; *Havlena et al., 2021*) influence the concentration of photosynthetic microorganisms in show caves. The presence of light may therefore be considered the main driver of photosynthetic microorganisms in show caves, overriding the role of other environmental parameters.

However, when considering the substrate effect, our analyses demonstrated distinctive substrate preferences for the three groups. Also, their substrate preference was influenced by light intensity, further contributing to a nuanced understanding of their ecological preferences.

Intriguingly, cyanobacteria did not exhibit any specific preference for a particular substrate and this pattern was confirmed also when accounting for the light intensity effect. This result suggests that cyanobacteria can successfully grow in show caves, irrespective of substrate type. We hypothesize that their success is mostly due to their high ecological plasticity (*Willis & Woodhouse, 2020*) allowing them to colonize almost every habitat on Earth, including deserts (*Wynn-Williams, 2000*), glaciers (*Zakhia et al., 2008*), hypersaline habitats (*Oren, 2010*), hot springs (*Ward, Castenholz & Miller, 2012*), and bones as well (*Marano et al., 2016*). In addition, they are complex prokaryotes, being able to form

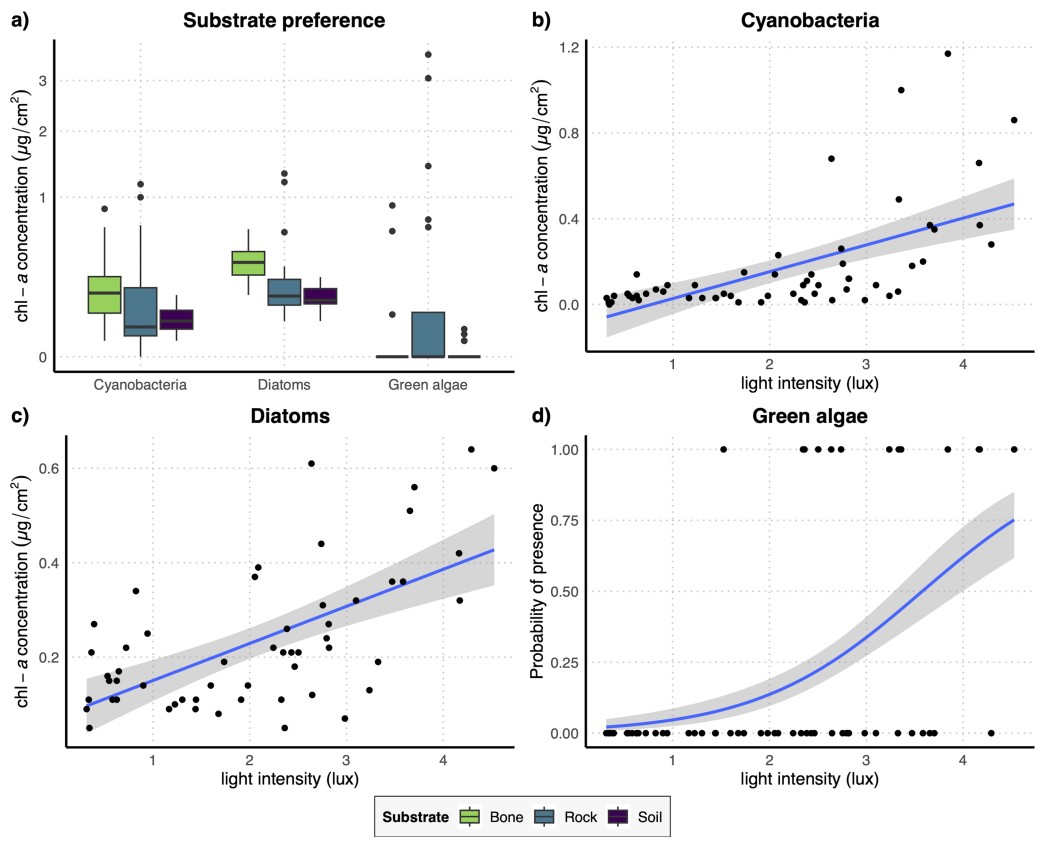

**Figure 3** (A) Boxplots representing the concentration of the three examined photosynthetic groups on the three different substrates obtained with the full dataset; (B–D) regression lines with confidence intervals representing the relation between the log-transformed light intensity and the concentration of cyanobacteria (B) and diatoms (C) and the probability of presence of green algae (D) obtained with the full dataset.

filament colonies and to adhere to different substrates thanks to the production of EPS (*Sciuto & Moro, 2015*).

In contrast, diatoms showed a significant affinity for bones over other substrates. This preference persisted even after accounting for the influence of light intensity, emphasizing that bones represent a preferential substrate for diatoms, overriding the effect of the main environmental driver, *i.e.,* light intensity, that significantly affected this groups in other studies (*e.g., Falasco et al., 2015*; *Piano et al., 2015*). Studies developed in forensic science demonstrated that diatoms can efficiently colonize recent bones and create biofilms on their surfaces, but species composition differs from other natural substrates, *e.g.,* cobbles in streams (*Marshall, Chraïbi & Morgan, 2023*). Regarding fossil bones, there is no evidence in literature of diatom colonization on this substrate, but the concentration of cyanobacteria on fossil deposits has been recorded and related to the elemental composition of this substrate (*Marano et al., 2016*). Although both calcite and bone apatite are dominated by calcium carbonate, the higher abundance of diatoms on bone substrates compared to rocks may be due to the different elemental composition of microelements found in fossil bones

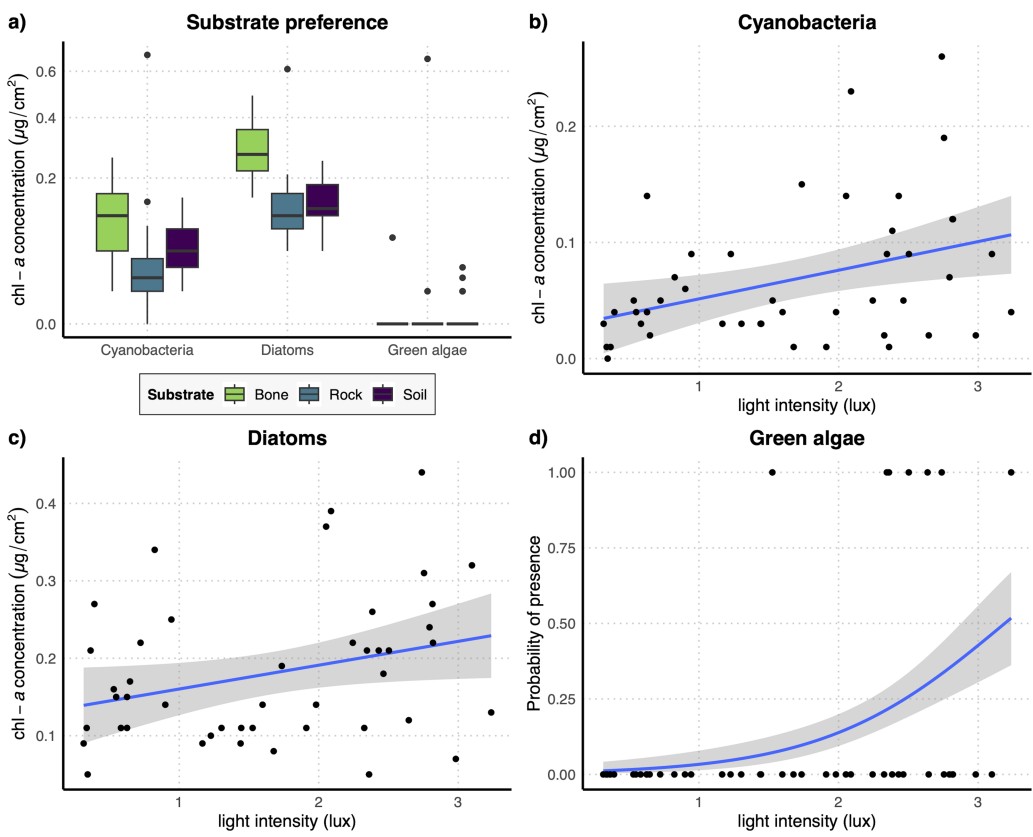

**Figure 4** (A) Boxplots representing concentration of the three examined photosynthetic groups on the three different substrates obtained with the restricted dataset; (B–D) regression lines with confidence intervals representing the relation between the log-transformed light intensity and the concentration of cyanobacteria (B) and diatoms (C) and the probability of presence of green algae (D) obtained with the restricted dataset.

derived from fossilization processes, *i.e.,* diagenetic change in bone structure and mineral composition (see *Xu & Poduska, 2014* for elemental spectra). For instance, nitrogen is found in high quantities on fossil bones (*Marano et al., 2016*), and especially phosphorus, a crucial nutrient that limits diatom concentration (*e.g., Winter & Duthie, 2000*; *Wilhelm et al., 2006*; *Brembu et al., 2017*; *Andersen et al., 2020*), is present in the fossiliferous sediments here examined in form of diagenetically produced phosphate minerals (*Rellini et al., 2021*; *Zunino et al., 2022*). In addition, other essential micronutrients, such as iron and manganese generated by oxidative processes, but also silica and zinc, are also present in the bone deposit, as indicated by *Rellini et al. (2021)* and *Zunino et al. (2022)*, likely enhancing the growth of diatoms on this substrate.

Furthermore, little breakages of bone surface, fracturing of bones and presence of abundant biological pitting (*i.e.,* predation marks and insect damages), together with a slight decalcification of bones due to dissolution, result in rough and porous surfaces (*Zunino et al., 2022*). These bone modifications increase the porosity of substrate compared to the surrounding rock, speleothems, and sediment, creating a surface that can promote

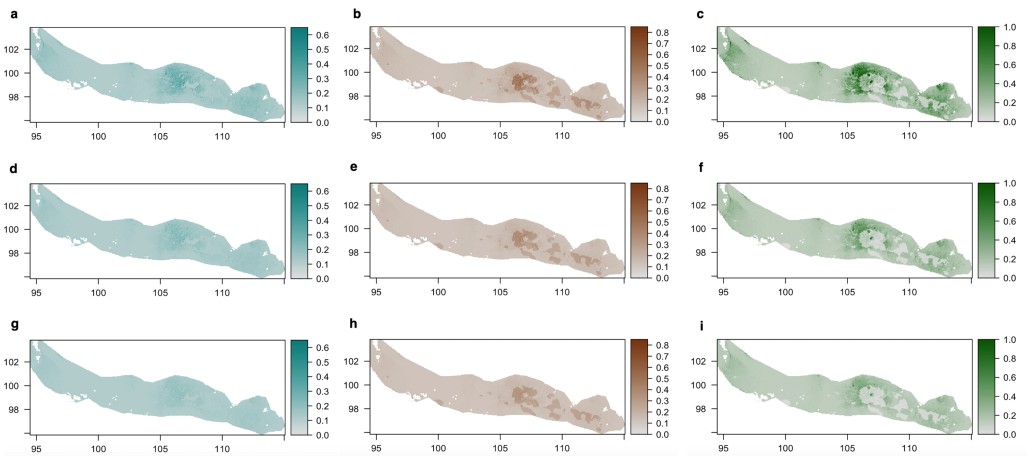

**Figure 5** **Density of cyanobacteria (A, D, G), diatoms (B, E, H) and green algae (C, F, I) on the 'Cimitero degli Orsi' expressed as chl-*a* concentration (μg chl-*a*/cm²) under different scenarios of light intensity: (A–C) present light intensity scenario; (D–F) 30% light intensity reduction; (G–I) 50% light intensity reduction.** Scenarios of light intensity were obtained by reducing the values of light intensity of 30% and 50% in the light raster, while predictions of the three groups were obtained by applying the coefficients of the statistical models to the three rasters of light intensity.

diatoms colonization as demonstrated by forensic studies on bones (*Marshall, Chraïbi & Morgan, 2023*) and synthetic polymers such as polyethylene (*Salimon et al., 2018*).

Based on our results, we may thus hypothesize that diatoms display significantly higher concentration on bone substrates due to higher availability of limiting nutrients, like zinc or silica, and to the presence of porosity and roughness of surfaces. To clarify the affinity between diatoms and fossil bones in caves, further studies are required involving analyses of microscopic structures and mineral contents of bones to understand as diagenetic processes have changed bone components and how these can affect the preference of diatoms for this substrate.

When analysing the full dataset, green algae demonstrated a substrate preference opposite to diatoms, with higher probability of presence on soil and rocks over bones. Considering that green algae are usually the first colonizers of the biofilm (*Mulec, Kosi & Vrhovšek, 2008*), we can hypothesise that they are outcompeted by cyanobacteria and especially diatoms on bone substrate when light intensity is high. The higher competition ability of diatoms and cyanobacteria was already demonstrated in previous research, where green algae were overridden on speleothems with seeping water, which represented a more favourable environment for their survival (*Piano et al., 2015*; *Piano, Nicolosi & Isaia, 2021*). More in detail, by analysing their metabolism and their genome, diatoms emerged as a highly competitive group that can survive in extreme environments and outcompete green algae under favorable environmental conditions (*Wilhelm et al., 2006*; *Behrenfeld et al., 2021*). However, green algae revealed a more nuanced relationship with the substrate type when adjusting our models for the confounding factor of light intensity. Specifically, it became evident that green algae did not exhibit a marked preference for any particular substrate, supporting their role as pioneer colonizers in show cave environments.

## CONCLUSIONS

In conclusion, our findings demonstrated that photosynthetic microorganisms can successfully colonize fossil bones in show caves when they are exposed to artificial lights and diatoms show higher concentrations on fossil bones than on other substrates. In light of these results, our study points out how illuminating fossil bones may jeopardize their conservation and ultimately their touristic and scientific value. Although cleaning strategies proved to be effective in removing *lampenflora* from speleothems (*Addesso et al., 2023b*), they cannot be adopted on paleontological remains because they may compromise their integrity. Reducing light intensity therefore represents the main strategy to diminish the concentration of photosynthetic microorganisms, as demonstrated by our models predicting *lampenflora* concentration under different light scenarios. Future studies should address the effectiveness of this management action by considering different strategies, such as reducing the intensity of light bulbs, avoiding direct lighting on paleontological remains, or installing lights with sensors near paleontological remains.

## ACKNOWLEDGEMENTS

The authors are grateful to Paolo Maschio and Nives Grasso for their help in collecting data for digital photogrammetry.

### Funding

This work was realized within the framework of the PRIN SHOWCAVE: A multidisciplinary research project to study, classify and mitigate the environmental impact in tourist caves'' - code 2017HTXT2R, funded by the Italian Ministry of Education, University and Research, and with the support of the projects PRIN DEEP CHANGE "Biodiversity conservation goes DEEP: integrating subterranean ecosystems into climate CHANGE agendas and biodiversity targets" (code PI: 2022MJSYF8, Stefano Mammola, funded by the Italian Ministry of Education, University and Research) and "National Biodiversity Future Center - NBFC" funded under the National Recovery and Resilience Plan (NRRP), Mission 4 Component 2 Investment 1.4 - Call for tender No. 3138 of 16 December 2021, rectified by Decree n.3175 of 18 December 2021 of Italian Ministry of University and Research funded by the European Union–NextGenerationEU (Project code CN_00000033, Concession Decree No. 1034 of 17 June 2022 adopted by the Italian Ministry of University and Research, CUP D13C22001350001). The grant of EP is co-financed by the PON "Research and Innovation" Programme (Axis IV "Education and Research for recovery"–Action IV.6 "Research contracts on Green themes"). The funders had no role in study design, data collection and analysis, decision to publish, or preparation of the manuscript.

### Grant Disclosures

The following grant information was disclosed by the authors:

The Italian Ministry of Education, University and Research: code 2017HTXT2R, code PI: 2022MJSYF8.
National Recovery and Resilience Plan (NRRP).
European Union–NextGenerationEU: Project code CN_00000033.
Italian Ministry of University and Research: CUP D13C22001350001.
PON ''Research and Innovation'' Programme.

## Competing Interests

The authors declare there are no competing interests.

## Author Contributions

- Elena Piano conceived and designed the experiments, performed the experiments, analyzed the data, prepared figures and/or tables, authored or reviewed drafts of the article, and approved the final draft.
- Marta Zunino performed the experiments, authored or reviewed drafts of the article, and approved the final draft.
- Giuseppe Nicolosi performed the experiments, authored or reviewed drafts of the article, and approved the final draft.
- Isabella Nicole Pisoni performed the experiments, analyzed the data, authored or reviewed drafts of the article, and approved the final draft.
- Alice Cimenti performed the experiments, authored or reviewed drafts of the article, and approved the final draft.
- Alberto Cina performed the experiments, authored or reviewed drafts of the article, and approved the final draft.
- Marco Isaia conceived and designed the experiments, performed the experiments, prepared figures and/or tables, authored or reviewed drafts of the article, and approved the final draft.

## Field Study Permissions

The following information was supplied relating to field study approvals (*i.e.*, approving body and any reference numbers):

Superintendency of Archaeology, Fine Arts, and Landscape for the Provinces of Imperia and Savona.

## Data Availability

The median values of chl-a concentrations of the three microorganism groups and the light intensity value and type of substrate where measurements were performed are available in the Supplemental File.

## Supplemental Information

Supplemental information for this article can be found online at http://dx.doi.org/10.7717/peerj.19622#supplemental-information.

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
