# Peer review of "Substrate type and light intensity determine lampenflora concentration on paleontological remains in show caves"

_PeerJ, doi:10.7717/peerj.19622_

## Round 0.1 · original submission · Major Revisions

The reviewers made useful and valuable suggestions to improve the paper.

Reviewer 1 ·

Basic reporting

No comment

Experimental design

No comment

Validity of the findings

No comment

Additional comments

The stydy explore lampenflora that develops on three different substrates (soil, rock and bones) by applying BenthoTorch and comparing the results obtained for cyanobacteria, green algae and diatoms. The results for these phototrophic groups were also related to light intensity and substrates using statistical analyses. The study is well designed and written. However, there are some suggestions that could be implemented.

Lines 75-81: Please mention only that lampenflora can cause aesthetic damage, physical and chemical deterioration and give references. These three processes are mentioned in many lampenflora papers, so perhaps they do not need to be explained.

You may add in the introduction, after the definition of lampenflora from which aspect is lampenflora studied worldwide and select some representative references, e.g. under the aspect of investigation and research, treatment and eradication, monitoring, etc.

Line 97: Please use another word instead of “scant”.

Lines 112-127: Please provide some information about the Toirano karst system first, then focus on the Toirano show cave that serves as a sampling site and write what environmental features the visited cave has

Line 138: Is "density" the right word? Or perhaps "concentration" is better? Please check through manuscript

Line 143: Italicize “in situ”

Line 164: Please use "software" instead of "environment"

Lines 241-246: Please separate the text clearly when writing about the dominance of a particular phototrophic group in terms of the number of sites where it was detected and their concentrations. For example: "In terms of the number of sites, diatoms and cyanobacteria were found…, while green algae... However, when the concentrations of these groups were examined with BenthoTorch, the highest concentration values were found in the green algae..."

Line 252: Please avoid repeating the same word in one sentence

Lines 257-258: The sentence “In other words, diatom concentration resulted to be higher on bone substrates compared to the other substrate types here examined” is a repetition of the statement from previous sentence. Perhaps you could merge them?

Lines 296-300: Please avoid repetition in these two sentences

Lines 300-308? Could the formation of EPS and its role contribute to the wide distribution of cyanobacteria and their development on different substrates and in different habitats?

Figures: I do not see full caption for Figure 3 and 4

Reviewer 2 ·

Basic reporting

The manuscript deals with a very interesting and actual topic, which is also very relevant from a practical point of view, providing important information, for instance, for the authorities. After minor revision, it is worth publishing.

Experimental design

In gerneral, the design of the experiment is basically good and meets the objectives. The statistical methods used are also appropriate.Tables and Figures are in the manuscript in sufficient number and quality. However, there are a few suggestions and questions:

-The type of fluorescent tubes used in the caves are missing from the description. However, in addition to the light intensity, its composition/spectrum is also important and can be quite important for algae growth. Without knowing this, it is very difficult to compare and use the results onbtained here with other research.
-Furthermore,the description of „soil” substrate type is very much missing in the manuscript (Line 112-131 contains the characterisation of the other two substrate types). What exactly does "bare soil" mean? What is its composition, what is its material structure, grain size, soil type, etc.?
-Why were only -30 and -50% light intensities modelled? what is the significance of these values (e.g. is there a plan to reduce the number of lamps at the official level)? Why no +light intensity were modelled?

Validity of the findings

The introduction is well written, precise, concise and logically structured. However, the hypotheses are missing. The modelling part is also missing from the objectives, it is only later that we find out that such studies have been done. It would be worth clarifying this at the outset.
DISCUSSION: The comparison of the 3 algal groups in terms of light intensity is missing, which tells us, for instance, that diatoms and cyanobacteria are also better competitor for light than green algae. There is a lot of literature on this subject, it would be worthwhile to refer to some basic articles (e.g. Reynolds, 1988).
Line 306-308: as only group-level measurements were made in the present study, species specificity is irrelevant at this stage. Within each algal group, "extreme" cases could be found. Or perhaps microscopic analyses have been done?
Line 321: As described by the authors, fossilization processes are very important e.g. for the availability of micronutrients. Is there any information or have any measurements been made, on the condition of the bones studied with respect to this process?

Additional comments

The title is a little bit misleading. One would expect that a lot of environmental parameters were investigated, but only light and substrate were choosen. For example, in a cave environment, the main environmental parameters for algal growth, besides light, may be humidity, airflow, temperature, pH..... Only a few of the main parameters have been investigated here- and in the case of light only the intensity-, so it is suggested to rephrase the title to better reflect the content of the article.

Reviewer 3 ·

Basic reporting

1. First letter of the "Lampenflora" should be in capital throughout. Similarly for "Showcave".
2. General English is satisfactory.
3. Why have authors only studied Lampenflora density and proliferation?
4. References are updated but sometimes not relevant to lampenflora. Mostly are related to cave culture and conservation. Authors need to revise this section.
5. Authors can add a table in the introduction showing comparative studies.

Experimental design

1. Authors have only used the spectra based on chlorophyll classes like chl-a, chl-c etc.. What about other parameters for density or variation in the sample? Any rock parameters or soil parameters?
2. Figure 5 is not clear. Density graph showing is quite similar.
3. What about the paleontological significance? Why have authors not measured marker-based studies to identify the age of the cyanobacteria, green algae and diatoms? Are they newly formed? Any fossilized spore or diatomaceous earth?
4. What about the proliferation of cyanobacteria, green algae, and diatoms? The authors have mentioned this, but no sufficient data is given.

Validity of the findings

1. The work is meaningful but the data is not sufficient.
2. Conclusion should highlight the impact and validity of the research but not others' research. This section should provide future directions.
3. The density dependent study is very regular but not very attractive.

Additional comments

Figures should be clear and the resolution needs to be improved.

---

## Round 0.2 · accepted · Accept

The initial draft of the paper was sent to three reviewers. Many of the comments of the reviewers overlapped. After revision, the revised paper was sent to two of the original reviewers, who both agreed that the authors had adequately addressed their comments and suggestions. I asked the third, more critical reviewer to review the revised paper, but the reviewer did not respond. However, I feel that the authors have adequately addressed the comments of this reviewer in their revised paper. As a result, I am satisfied that the paper can now be accepted for publication.

Reviewer 1 ·

Basic reporting

.

Experimental design

.

Validity of the findings

.

Additional comments

The authors have responded to all comments made in the first round of reviews. This is a good paper, I have no further comments.

Reviewer 2 ·

Basic reporting

The manuscript has been significantly improved. The completions, modifications and responses are adequate. In its current form it is now fully suitable for publication without further modification.

Experimental design

-

Validity of the findings

-

Additional comments

-